# Updates in the Diagnosis and Management of Linear IgA Disease: A Systematic Review

**DOI:** 10.3390/medicina57080818

**Published:** 2021-08-12

**Authors:** Leah Shin, Jeffrey T. Gardner, Harry Dao

**Affiliations:** 1Loma Linda University School of Medicine, Loma Linda, CA 92708, USA; Lshin@llu.edu; 2Department of Dermatology, Loma Linda University School of Medicine, Loma Linda, CA 92708, USA; JeffreyGardner@llu.edu

**Keywords:** linear IgA bullous dermatosis, autoimmune diseases, immunoglobulin A, fluorescent antibody technique, rituximab, etanercept, omalizumab

## Abstract

*Background and Objectives:* Linear IgA disease (LAD) is a rare autoimmune blistering disease with linear IgA deposits along the basement membrane zone. Direct immunofluorescence remains the gold standard for diagnosis, but other diagnostic measures reported in recent literature have proven useful in the setting of inconclusive preliminary results. Dapsone is a commonly used treatment, but many therapeutic agents have emerged in recent years. The objective of this study is to provide a comprehensive overview of updates on the diagnosis and management of LAD. *Materials and Methods:* A literature search was conducted from May to June of 2021 for articles published in the last 5 years that were related to the diagnosis and management of LAD. *Results:* False-negative results in cases of drug-induced LAD and the presence of IgG and IgM antibodies on immunofluorescence studies were reported. Serration pattern analysis has been reported to be useful in distinguishing LAD from sublamina densa-type LAD. Rituximab, omalizumab, etanercept, IVIg, sulfonamides, topical corticosteroids, and others have been used successfully in adult and pediatric patients with varying disease severity. Topical corticosteroids were preferred for pediatric patients while rituximab and IVIg were used in adults with recalcitrant LAD. Sulfonamides were utilized in places without access to dapsone. *Conclusion:* In cases where preliminary biopsy results are negative and clinical suspicion is high, repeat biopsy and additional diagnostic studies should be used. Patient factors such as age, medical comorbidities, and disease severity play a role in therapeutic selection.

## 1. Introduction

Linear IgA disease (LAD) is an autoimmune mucocutaneous disease characterized by linear deposits of IgA at the basement membrane zone on immunopathology [1]. It is also known as linear IgA bullous dermatosis (LABD), but LAD is preferred because it is inclusive of patients without bullous lesions [2]. In the pediatric population, it is known as chronic bullous disease of childhood (CBDC). Direct immunofluorescence (DIF) remains the gold standard for diagnosis in both adult and pediatric populations, but there have been cases of false-negative results in drug-induced LAD [3,4]. Management of this relatively rare disease process varies throughout the literature. Dapsone is the most commonly used therapeutic agent, but its potential side effects such as hemolysis, agranulocytosis, and methemoglobinemia necessitate the use of other treatment modalities. Monitoring for dapsone adverse reactions can be cumbersome as well. Numerous other treatments have been reported to be effective in the treatment of LAD, including topical corticosteroids, tetracyclines, dicloxacillin, oxacillin, erythromycin, sulfonamides, nicotinamide, rituximab, omalizumab, methotrexate, cyclosporine, etanercept, and intravenous immunoglobulin (IVIg) [1,5,6,7,8,9,10,11,12,13,14,15,16,17,18,19,20,21]. This review provides updates on the diagnosis of LAD and emerging treatment modalities in order to assess their utility in the management of this disease.

## 2. Materials and Methods

A literature search was conducted from May to June of 2021 using the online database PubMed. To maximize results, a broad search term “linear igA” was used. Titles and summaries of articles were screened for relevance to LAD diagnosis and treatment, followed by the assessment of abstracts and full-text manuscripts. Information on the diagnosis and treatment of LAD were extracted by two independent reviewers. Only articles published between 2016 and 2021 with full-text access and updates to LAD diagnosis and management were included in this review.

## 3. Results

### 3.1. Literature Search

A preliminary search yielded 401 articles published between 2016 and 2021 that were related to “linear igA”. The titles, summaries, and abstracts were screened for relevance to the topic, leaving a total of 65 articles. These studies were assessed further by reading the abstract or full text. Articles without full-text access or those that were unrelated to diagnosis or management were not retrieved. This left 30 studies that met the inclusion criteria, in addition to 2 articles recommended by peer reviewers (Figure 1).

### 3.2. Diagnosis

Recent reports show that conventional diagnostic studies may not be the most accurate for drug-induced LAD. One case of vancomycin-induced LAD in a patient with renal insufficiency initially showed a negative DIF, but upon repeat biopsy, DIF result was positive [3]. This case highlights the importance of repeat DIF if clinical suspicion is high and if patients have immune dysregulation such as renal insufficiency, which can alter immunofluorescence studies. In another case of vancomycin-induced LAD, despite positive DIF results, indirect immunofluorescence (IIF) was negative unless the serum was co-incubated with the offending agent, vancomycin [4]. A unique flame figure formation with numerous eosinophils on hematoxylin and eosin (H&E) has also been reported in drug-induced LAD [22].

In situations where similar conditions such as when sublamina densa-type LAD is in the differential, IIF is often used to aid in the diagnosis. Unfortunately, up to 30% of patients with LAD can have a negative IIF. If this is the case, serration pattern analysis can help distinguish between the two. LAD has an “n-serrated” pattern versus the “u-serrated” pattern of sublamina densa-type LAD [23]. It is significant to differentiate between the two because treatment response can vary depending on the diagnosis. In one study, three out of four patients with sublamina densa-type LAD did not respond well to dapsone and had to be placed on combination therapy with IVIg, doxycycline, rituximab, mycophenolate mofetil (MMF), as well as oral and topical corticosteroids [24]. One point worthy of note is the term IgA epidermolysis bullosa acquisita (EBA), which is considered arcane because it does not incorporate the immunologic or molecular qualities of the disease [2].

Though LAD is typically associated with IgA on immunofluorescence testing, there are exceptions. IgG can be seen with IgA on DIF, which some call linear IgA/IgG bullous dermatosis (LAGBD). In a pediatric patient with both IgA and IgG positivity on DIF, dapsone alone was not enough to control ocular involvement, prompting the use of oral prednisone and corticosteroid eyedrops [25]. IgM has also been seen, albeit rarely, with IgA and IgG in a linear fashion in the basement membrane zone, although IIF results were negative for IgA and IgG [26]. Another study found that around 50% of patients with LAD were positive for IgG on enzyme-linked immunosorbent assay (ELISA) or immunoblotting while being negative for linear IgG on DIF [27].

### 3.3. Treatment

In recent years, rituximab, IVIg, and others have been increasingly utilized and reported due to the adverse effects, lack of access, and treatment failure of dapsone (Table 1). These agents were often used in combination with systemic corticosteroids. There are only a few published cases that used omalizumab and etanercept, therefore data are limited on the efficacy of these agents as a treatment for LAD. In the pediatric population, management tended to be more conservative with systemic and topical corticosteroids or nicotinamide. Dosing regimens, time to clearance, and relapse rates varied throughout this literature review.

#### 3.3.1. Rituximab

Rituximab is an anti-CD20 antibody approved by the FDA for pemphigus vulgaris and used off-label for other autoimmune bullous diseases. Four articles cite rituximab as an effective additional agent for severe, recalcitrant LAD. Three of the articles used two infusions of 1 g, 2 weeks apart, for two cycles [8,9,12]. One used a dosage of 375 mg/m^2^ weekly, for a month, with mycophenolate mofetil 500 mg BID (bis in die) after the lack of response to prednisone [11]. In a case with severe ocular involvement, rituximab in combination with IVIg (2 g/kg/cycle over 3 days, for two cycles) was used [12]. Other sources added rituximab to a regimen of dapsone, prednisone, doxycycline, and MMF due to the lack of response. Time to clearance ranged from 7 weeks to 20 months. Relapses occurred in two out of five patients with LAD, 6 and 9 months after clearance [8,11]. No significant side effects were noted in patients with LAD in any of the articles. However, it may be of benefit for patients to still be clinically followed due to rituximab’s immunosuppressive features that place them at greater risk of infections.

#### 3.3.2. Methotrexate

One case of LAD associated with chronic ulcerative proctitis resulted in clinical improvement only after the patient’s proctitis was treated with methotrexate (22.5 mg/week) and mesalamine [15]. The patient failed to respond to previous therapy with dapsone, niacinamide, doxycycline, prednisone, intravenous methylprednisolone, topical corticosteroids, and mesalamine monotherapy.

#### 3.3.3. Sulfonamides

Sulfamethoxypyridazine and sulfasalazine are sulfonamides that have been reported as effective treatments for LAD, with minimal risk [6,28]. Since dapsone and some sulfa drugs are difficult to obtain in countries such as China, oral sulfasalazine at 40–60 mg/kg/day was given to a 17-year-old boy in China along with oral sodium bicarbonate to prevent the formation of urinary tract crystals for two months [6]. Another study in Kuwait transitioned a 7-year-old from dapsone to sulfamethoxypyridazine (250 mg/day to 1 g/day for 60 months) because of persistent hemolysis [28]. No relapses or side effects were reported in both patients.

#### 3.3.4. IVIg

IVIg is used in immune-mediated diseases, including autoimmune dermatologic conditions. Regarding LAD, three studies used IVIg either as a primary or adjunct treatment if the disease was severe and/or therapy-resistant [12,19,28]. The dose used was 2 g/kg over a few days, without relapses. Scarpone et al., found that the common side effects of IVIg were headache, abdominal pain, nausea, vomiting, diarrhea, and chest pain in patients with dermatomyositis, vasculitis, vasculopathy, and pemphigus vulgaris. However, no side effects were reported in the patient with LAD. Time to clearance varied from 7 months to 96 months, with a median of 7 cycles.

#### 3.3.5. Topical Corticosteroids

Corticosteroids are used in a wide variety of dermatologic conditions. Although not a new treatment for LAD, topical corticosteroids have been popular in the pediatric population for its favorable side effect profile. In a case of uncomplicated CBDC upon delivery with limited mucous membrane involvement, topical treatment with betamethasone valerate 0.05% at day 4 of life resulted in resolution at day 21 of life [21]. In a 6-year-old female with partial glucose-6-phosphate dehydrogenase deficiency, topical methylprednisolone 0.1% twice daily (vehicle not specified) and clobetasol shampoo resulted in resolution within a month [20]. In an adult case of vancomycin-induced LAD, vancomycin cessation and triamcinolone 0.1% ointment alone were enough to resolve bullae [3]. It is unclear as to how long after clinical improvement the patient continued to use topical corticosteroids. Two other cases of mild CBDC showed clinical improvement after 2.5 months of unspecified, mid-potency topical corticosteroid therapy [30]. One of these cases used topical corticosteroids in combination with clarithromycin at 30 mg/kg/day for 1 month. No cases of topical corticosteroid use reported any side effects.

#### 3.3.6. Systemic Corticosteroids

Systemic corticosteroids were commonly paired with dapsone or IVIg in the treatment of LAD. Prednisolone was used in pediatric patients at a dose of 0.5–1 mg/kg/day, while prednisolone or prednisone were used at doses of up to 1 mg/kg/day in adults [16,19,25,28,29]. In Nanda et al., prednisolone in tapering doses was added to the dapsone regimen if response was slow. It was also added to IVIg in moderate to severe cases if dapsone was contraindicated. In a severe case of CBDC requiring inpatient admission, a combination of dapsone, prednisolone, and cyclosporine of unspecified dose resulted in a well-controlled disease [16]. The sequence of medication initiation and therefore the role of prednisolone versus cyclosporine is unclear in this case. Another severe case of CBDC required dapsone at 2 mg/kg/day along with prednisolone and corticosteroid eye drops for treatment-resistant conjunctival involvement [25]. In adults, the use of prednisone alone or in combination with dapsone has resulted in variable success, as reported by Machado et al. Despite high doses of dapsone (100 mg/day reduced to 50 mg/day due to hemolysis) and prednisone (1 mg/kg/day) in one patient, response remained poor. After 5 years of chronic corticosteroid use, she developed Cushingoid facies, arterial hypertension, glaucoma, and osteoporosis. The second patient had the disease completely under control and suspended medication use after 2.5 years of dapsone (100 mg/day reduced to 50 mg/day due to hemolysis) and prednisone (0.6 mg/kg/day). The third patient showed improvement after 1 month of prednisone at 20 mg/day, with relapse after self-initiated cessation of prednisone. Corticosteroids were tapered as clinical improvement was obtained.

#### 3.3.7. Nicotinamide

Nicotinamide inhibits inflammatory pathways such as leukocyte chemotaxis, lysosomal enzyme release, mast cell degranulation, and more [7]. There is one article reporting the sole use of nicotinamide to treat CBDC in a 22-month-old Chinese patient. The patient had an 11-day history of lesions after an insect bite, after which nicotinamide at 300 mg/day was initiated. The lesions were completely resolved after 7 days of treatment, without any side effects, and she had no relapses by her 10-month follow-up.

#### 3.3.8. Amoxicillin-Clavulanate

One case of neonatal CBDC in a 5-day-old infant without mucosal involvement showed a prompt resolution of the disease after a 7-day amoxicillin-clavulanate infusion of unspecified dose [31]. Amoxicillin-clavulanate was started due to maternal fever and the presence of lesions. However, maternal and neonatal microbiological samples were negative. No relapses were reported after the 20-month follow-up.

## 4. Discussion

The management of LAD has centered around DIF for diagnosis and around dapsone for treatment. Additional diagnostic studies include IIF for IgA anti-basement membrane zone antibodies and H&E stains showing subepidermal blisters with predominant neutrophils, with or without eosinophils later in the disease process [1]. Repeat biopsy is indicated if clinical suspicion is high in patients with immune dysregulation [3]. Recent literature has shown that serration pattern analysis may be a useful tool for distinguishing between similar presentations such as sublamina densa-type LAD because treatment response will vary depending on the diagnosis [23,24]. In cases of drug-induced LAD, co-incubation with the offending agent has been reported to be helpful in the prevention of false negative IIF results [4]. In addition to IgA, there are cases of IgG and IgM on DIF [25,26]. The presence of IgG was associated with ocular involvement and required treatment with both dapsone and corticosteroids in one pediatric case [25]. Due to the small sample size, it is unclear how the presence of multiple antibodies affects clinical management. A wide array of diagnostic testing options should be considered if clinical suspicion is high for LAD.

There are many different treatment options for LAD, underscoring the challenge of treating this disease with variable clinical presentations and underlying causes. Due to dapsone’s side effects, its difficult accessibility in some countries, and the lack of response, other treatments have been utilized. Rituximab and IVIg were typically used for more severe and therapy-resistant cases, while omalizumab was used when there seemed to be a heavy eosinophil involvement. Etanercept was used in vancomycin-induced LAD presenting as TEN, with prompt resolution after one dose. Associated conditions such as ulcerative colitis should be investigated and treated if clinical response to traditional therapy is poor. In the pediatric population where the side effect profile is a priority, sulfonamides, topical corticosteroids, and nicotinamide were preferable. Sulfonamides were used in areas with less access to dapsone, topical corticosteroids were used in newborns and young children, while nicotinamide was used in a toddler. All achieved complete remission without any noted side effects or relapses. Newborns, in particular, should be assessed for precipitating factors that lead to CBDC presentation such as IgA autoantibodies in breast milk, even if mothers are asymptomatic [32]. Many reports did not mention dosage, presence of relapse, side effects, or time to clearance, thus limiting the scope of this review. Another limitation of this review was that “CBDC” was not searched as it was assumed that “linear IgA” would have been a keyword or searchable term in the studies. Although dapsone is most often recommended to patients with LAD, other treatment modalities can be considered based on comorbidities and the response to therapy.

## 5. Conclusions

LAD management has relied on DIF for diagnosis and on dapsone for treatment. A literature search for articles published in the last 5 years has shown that other modalities, including rituximab, omalizumab, etanercept, IVIg, topical corticosteroids, among others have been used successfully in adult and pediatric patients with varying severity of disease. Patient factors such as age, medical comorbidities, and disease severity play a role in therapeutic selection. As in most patient cases in Dermatology where preliminary biopsy results are negative but clinical suspicion is high, a repeat biopsy and additional diagnostic studies should be considered, especially in LAD. 

## Figures and Tables

**Figure 1 medicina-57-00818-f001:**
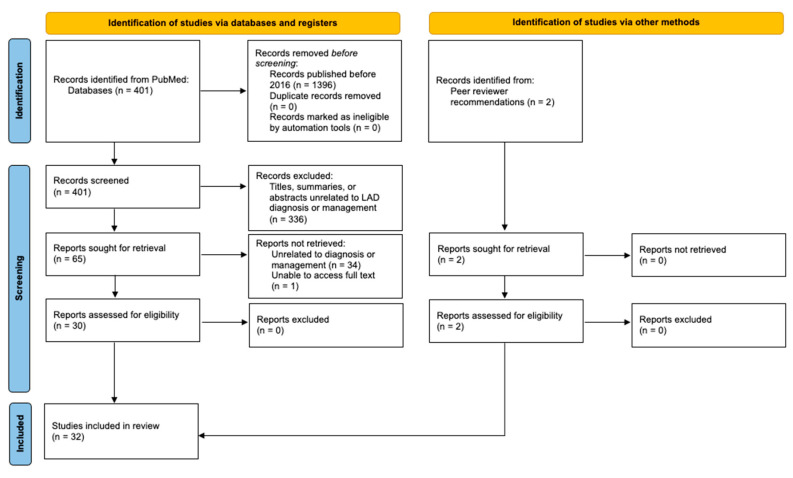
Flow diagram of the systematic review.

**Table 1 medicina-57-00818-t001:** Review of alternative therapeutic agents documented in the literature from 2016 to 2021.

Drug	Dose	Age (Years)	Drug-Induced LAD (Y/N)	Time to Clearance (yrs/m/w/d)	Relapse (Y/N)	Side Effects of Treatment
Rituximab + prednisone + MMF [11]	Rituximab: 375 mg/m^2^ weekly × 4 w Prednisone: 0.5 mg/kg (80 mg) tapered over 1 m MMF: 500 mg	43	N	1 m	Y, 9 m later	NS
Rituximab + dapsone + topical corticosteroids + MMF [8]	Rituximab: 2 infusions of 1 g 2 w apart × 2 Dapsone: unspecified Topical corticosteroids: unspecified MMF: 3 g/d to 500 mg/d	35	Unknown	14 m	Y, 6 m later	NS
Rituximab + dapsone + IVIg [9]	Rituximab: 1 g × 2 cycles Dapsone: 100 mg to 50 mg IVIg: 2 g/kg/cycle	33	N	7 w	N	None
Rituximab + dapsone + prednisone + doxycycline + MMF [8]	Rituximab: 2 infusions of 1 g 2 w apart × 2 Dapsone: 200 mg/d Prednisone: 0.1 mg/kg/d Doxycycline: 200 mg/d MMF: 1 g/d	30	Unknown	20 m	N	NS
Rituximab + IVIg [12]	Rituximab: 2 infusions of 1 g 2 w apart IVIg: 2 g/kg/cycle divided over 3 d × 2	21	Unknown	17 m follow-up showed improvement in visual acuity and less conjunctival cicatrization	N	None *
Omalizumab	Dose regimen 1: Subcutaneous 300 mg every m × 6 m [13] Dose regimen 2: Subcutaneous 300 mg every m × 3 m [14]	55 40	N Unknown	3 w 4.5 m	Y, within 1 m of cessation N	NS NS
Etanercept [17]	50 mg × 1	65	Y	4 d	NS	None
Methotrexate + mesalamine ** after high dose prednisone + IV methylprednisolone [15]	Methotrexate: 22.5 mg/w Mesalamine: unspecified Prednisone: unspecified Methylprednisone: 3 d course	58	N	NS, but near clearance achieved	NS	NS
Sulfasalazine [6]	40–60 mg/kg daily	17	N	2 m	N	None
Sulfamethoxypyridazine [28]	250 mg–1 g/d	7	N	60 m	N	NS
IVIg + prednisolone [19]	IVIg: 2 g/kg or 0.4 g/kg for a median of 6 doses over 2–5 d + Prednisolone: 5–10 mg/d	Unspecified, range of 64–84	Unknown	NS	N	None ***
IVIg + prednisolone [28]	IVIg: 2 g/kg/cycle × 8 Prednisolone: 0.5–1 mg/kg/d	13	N	30 m	N	NS
IVIg [28]	2 g/kg/cycle × 8 cycles	9, 1.7	N	96 m, 7 m	N	NS
IVIg + prednisolone + clarithromycin [28]	IVIg: 2 g/kg/cycle monthly × 5 Prednisolone: 1 mg/kg/d Clarithromycin: 30 mg/kg/d over 3 doses	1	N	6 m	N	NS
Dapsone + prednisone [29]	Dapsone: 50 mg/d Prednisone: 0.6 mg/kg (60 mg/d)	51	N	2.5 yrs	N	NS
Dapsone + prednisone [29]	Dapsone: 50 mg/d Prednisone: 1 mg/kg/d (60 mg/d) tapered to 10 mg/d	44	N	Clearance not achieved	Clearance not achieved	Glaucoma, arterial hypertension, osteoporosis, Cushingoid facies
Prednisone [29]	20 mg/d monthly dose tapering	30	Unknown	NS	Y	NS
Dapsone + prednisolone [28]	Dapsone: 1–2 mg/kg/d Prednisolone: 0.5–1 mg/kg/d in tapering doses	13, 9, 7, 1	N	60 m, 156 m, 108 m, 24 m	N	NS
Oral corticosteroids [30]	Unspecified	8	NS	4 m	NS	NS
Oral prednisolone + corticosteroid eye drops [25]	0.5 mg/kg/d	7	NS	NS, but had eyelid adhesion despite clinical control	NS	NS
Dapsone + prednisolone + cyclosporine [16]	Dapsone: 0.5–2 mg/kg; Prednisolone: 0.5–1 mg/kg Cyclosporine: dose unspecified	5	N	NS, but is well-controlled	NS	NS
Topical triamcinolone + vancomycin cessation [3]	Triamcinolone 0.1% ointment	74	Y	NS, but clearance was achieved	NS	NS
Topical methylprednisolone + clobetasol shampoo [20]	Methylprednisolone: 0.1% BID × 8 w	6	N	5–8 w	N	NS
Topical corticosteroids + clarithromycin [28]	Topical corticosteroids: mid-potency, unspecified Clarithromycin: 30 mg/kg/d over 3 doses × 1 m	5	N	2.5 m	N	NS
Topical corticosteroids [28]	Mid-potency, unspecified	4	N	2.5 m	N	NS
Betamethasone valerate [21]	0.05%	4 d	N	21 d	NS	NS
Nicotinamide [7]	300 mg/d	22 m	N	7 d	N	None
Amoxicillin-clavulanate [31]	Infusion × 7 d	7 d	N	7 d	N	NS

yrs, years; m, months; w, weeks; d, days; Y, Yes; N, No; NS, not stated. * While the patient with LAD did not experience side effects from Rituximab + IVIg, one patient diagnosed with MMP experienced pneumonia and life-threatening septicemia 3 weeks after RTX infusions, which was successfully treated with very aggressive intravenous antibiotic therapy. ** Patient was also being treated for chronic ulcerative proctitis. *** While the patient with LAD did not experience side effects from IVIg, patients in the study with pemphigus vulgaris, vasculitis, vasculopathy, and dermatomyositis experienced headache, abdominal pain, gastrointestinal upset, and chest pain.

## Data Availability

No new data were created or analyzed in this study. Data sharing is not applicable to this article.

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
