# Peer review of "Updates in the Diagnosis and Management of Linear IgA Disease: A Systematic Review"

_medicina, 2021, doi:10.3390/medicina57080818_

Round 1

Reviewer 1 Report

It is a concise and well-written article. For the sake of clarity, in section 3.3 I would mention topical treatment (topical steroids) before systemic ones.

Author Response

Thank you for your suggestion. We have switched the order in the treatment section to mention topical corticosteroids before systemic corticosteroids.

Reviewer 2 Report

This is a comprehensive and balanced summary and guidance for a rare but important disease.  I have 2 comments:  1)  The term corticosteroid is used only for systemic use, and steroid is used for topical use.  The term corticosteroid should be use for all modes of use, steroid is simply a vague jargon.  2)  There is an excellent paper on the transference of IgA from a mother to baby via breast milk.  I think a short reference is worthwhile for the review submitted.  [Neonatal linear Ig A...JAMA Dermatol. doi:10.1001/jamadermatol.2021.2392    Published online July 14, 2021.

Author Response

Thank you for your suggestions.

1) We have changed steroid to corticosteroid throughout our paper. 

2) This is a very interesting article, thank you for pointing it out. We have added it to our discussion (lines 282-284).

Reviewer 3 Report

The comments to the authors:

This is an extensive review article for linear IgA bullous dermatosis (LABD)/linear IgA dermatoses (LAD), particularly for the diagnoses and treatments.  This review should be important for the better understanding the current situation in LAD.  However, I have a number of concerns and comments, which are described below. 

(1) First of all, there is a discussion for the nomenclature issues of this skin condition (Hashimoto et al. JAMA Dermatol. 2021 Jun 23. doi: 10.1001/jamadermatol.2021.0761.).  The major point is that the name of LAD may be better to be used, rather than LABD, because some of the patients with this condition may not show bullous lesions.  This point should be discussed, and the term of LAD may be used in this review article, rather than LABD.

(2) Related to the comment (1), the second point for the nomenclature of LAD is which of the two terms; sublamina densa-type LAD or IgA epidermolysis bullosa acquisita (EBA), should be used for the LAD cases with IgA reactivity with dermal side of the split skin in indirect immunofluorescence and/or with type VII collagen (Hashimoto et al. JAMA Dermatol. 2021 Jun 23. doi: 10.1001/jamadermatol.2021.0761.).  Because the term IgA EBA is not generally used, and because the term EBA itself is arcane and old fashioned, the term sublamina densa-type LAD, rather than IgA EBA, is suitable for this condition.  This point should be discussed, and the term IgA EBA may be replaced by the term sublamina densa-type LAD throughout the manuscript.

(3) This review included cases with phenotype of mucous membrane pemphigoid (MMP) and IgA immunoreactivity as LAD.  However, these cases should be considered MMP with predominant IgA antibodies, but not LAD.  Therefore, the authors should remove these MMP cases from this review.

(4) The table 1 is too large.  All the contents in this table should be summarized, and should be shown as a much smaller table.

(5) In the abstract section, the important findings for the diagnoses and treatments in this review article should be described in more detail.

(6) There are duplicated descriptions between the introduction section and sections of the results and discussions.  Such duplicated descriptions should be removed.

(7) There is/are only one or very few paper(s) for the efficacy of etanercept and omalizumab in the treatments of LAD.  Therefore, the sections for etanercept and omalizumab may be removed, and these issues may be stated shortly in the introduction of the treatment section.  

(8) In the results section, as well as the figure 1, the literature search should be started after 2016, and only the 401 articles, rather than 1797 articles, should be selected for the first candidate papers, because the authors aimed to examine the cases after 2016.

(9) The two sentences in the lowest column in the figure 1 seem to be the same.

(10) The explanations for “Y” and “N” are missing in the footnote for the table 1.

(11) Although the authors examined 30 previous papers for this review, there are only 30 papers in the reference list.  Are all of the 30 papers in the reference list the papers, which the authors examined?

(12) English needs improvements in some places.  For example, the phrases “exclude many patients from being potential candidates” and “This review identifies updates” in the introduction section seem to be inappropriate.  The word “oftentimes” in the treatment section may be changed to “often”.  The full spell for “BID” may be need in the treatment section.

Author Response

We appreciate your thorough review of our paper.

1) Thank you for pointing this out. We have addressed this point in lines 32-33 and changed LABD to LAD. 

2) Your point was addressed in the second paragraph of 3.2 (lines 79-91)

3) The mention of MMP in the table was clarified (Row 6). The patient who had side effects in the Rituximab + IVIg study was not a patient with LAD, but one with MMP. We changed the side effects referring to the patient with MMP to None and added a footnote at the end of the table explaining that there was a patient with MMP on the same treatment regimen that developed side effects (Lines 114-116). 

4) The font was made smaller, abbreviations used, and some rows combined to make the table smaller. We thought it would be important to include the original columns and treatments in order to better visualize the case by case nature of these treatments. 

5) We have added more detail regarding management of LAD into the Results portion of the abstract (Lines 19-22).

6) The redundant sentence was removed in the second paragraph of the discussion (Lines 268-271).

7) Sections for omalizumab and etanercept were removed (lines 136-155) and a short sentence regarding these treatments was added to the beginning of the treatment section (lines 107-109).

8) Results section and figure 1 were changed to reflect a literature search after 2016 that started with 401 articles instead of 1797 (lines 58-60, 66).

9) Removed the duplicate sentence in figure 1.

10) Added the explanations for Y and N at the end of table 1 (line 113)

11) Yes, the 30 papers (now 32 after reviewer suggestions) that we examined were the papers included and referenced in this review.

12) These phrases were fixed: "excludes many patients from being potential candidates" was changed to "side effects ... necessitate using other treatment modalities" (lines 39-40), "This review identifies updates" was changed to "provides updates" (line 45), "oftentimes" to "often" (line 105), and the BID acronym was explained in parenthesis after it was used in line 126. 

Round 2

Reviewer 3 Report

None.